# Local Territorial Practices Inform Co-Production of a Rewilding Project in the Chilean Andes

Matías Guerrero-Gatica [1,2,3,*], Tamara Escobar Reyes [3,4], Benjamín Silva Rochefort [3,5], Josefina Fernández [3,5], Andoni Elorrieta [3,5] and Meredith Root-Bernstein [2,3,6,7]

1   Departamento de Geografía, Universidad de Chile, Santiago 8331051, Chile
2   Instituto de Ecología y Biodiversidad (IEB), Barrio Universitario, Concepción 4070374, Chile
3   ONG Kintu, Ñuñoa, Santiago 7800004, Chile
4   Social Sciences Faculty, Universidad de Chile, Ñuñoa 7800020, Chile
5   Department of Biological Sciences, Pontificia Universidad Católica de Chile, Santiago 8331150, Chile
6   UMR Centre d'Écologie et Sciences de la Conservation, CNRS, Muséum National d'Histoire Naturelle, 75005 Paris, France
7   Center of Applied Ecology and Sustainability CAPES-UC, Pontificia Universidad Católica de Chile, Santiago 8331150, Chile
*   Correspondence: mguerrerog@ug.uchile.cl

**Abstract:** Co-production of conservation projects is favored by incorporating local ecological knowledge into project design and implementation. Using a mixed method approach, we asked how the territorial practices and knowledge of cowboys and livestock farmers inform their attitudes to this proposed project. We predicted that cowboy territorial practices would be reduced in diversity compared to the past, and that this may be associated with a reduction in coping or adaptation capacity in the face of environmental challenges. We further predicted that due to growing environmental and social pressures reducing traditional livelihood opportunities for this group, they are likely to see the guanaco reintroduction project in a conflictual and negative light. We additionally predicted that they would perceive local carnivorous species in a conflictual and negative way. We found that territorial practices among the sample had indeed decreased in diversity. The sample coped with changing socio-ecological conditions by taking up other jobs. However, we also found that they had majority favorable views on the guanaco reintroduction project. Yet their knowledge of current guanaco behavior led them to believe that the project would fail. However, they also observed that pumas and condors changed their behaviors. We suggest that there are opportunities to co-produce knowledge about the possibility of flexible and adaptive guanaco behavior, which may lead to restoration and create more sustainable future scenarios, by engaging with the territorial practices and local ecological knowledge of cowboys and livestock farmers.

**Keywords:** restoration; reintroduction; territory; semi-arid ecosystems; mountain; local knowledge

## 1. Introduction

Co-production of conservation projects is an increasingly recognized approach for incorporating local ecological knowledge into project design and implementation [1–5]. In a co-production approach, projects incorporate local ecological knowledge, experience, interests and preferences into their design from the early stages [4,6]. This is expected to facilitate appropriate recognition of local knowledge, concerns, and needs, and to enhance project relevance to the local community. Explicit co-production approaches have not been trialed in rewilding contexts, partly because rewilding projects are frequently situated in areas of human land abandonment, or on private lands without resident communities, for reasons which may be either ideological or pragmatic [7–9]. However, other approaches to rewilding, for example in South America, explicitly embrace rewilding in landscapes with human livelihoods [8,10]. One aspect that clearly needs to be addressed in such contexts,

and which has received attention, is working with communities to reduce human–wildlife conflict, introduce or reinforce practices of coexistence, and strengthen pro-conservation and pro-wildlife values [10,11]. Another important angle of rewilding in livelihood landscapes is understanding how livelihood practices contribute to forming the territory in question, and how rewilding and territorial practices may interact to alter human and non-human species' territorial relationships.

By territory we mean a space that is socially produced, constructed, and interpreted [12]. Territories are where space has been socially constructed as meaningful places [13–18]; they are the sociocultural appropriation of physical space [19,20]. The dynamic expression of diverse structures, practices and relations is what we call "territoriality" [12,19,20]. To fully grasp the social as well as ecological or biophysical aspects of territorial uses and practices requires a transdisciplinary perspective [21]. Territoriality provides a powerful theoretical and methodological approach for understanding sociocultural aspects of local communities in rewilding contexts. A territorial approach calls for a deeper understanding of local ecological knowledge and how it is enmeshed with practice and mobilized during intercultural collaboration [22,23]. It also emphasizes that knowledge and practice are together intimately linked to the land, both creating the physical substrate of territory and in turn shaping the conditions for further practice and learning [24–27]. Since rewilding projects inherently aim to alter landscape processes, dynamics, and structures, a territorial approach helps us to understand how these changes will affect livelihoods, alter learning and knowledge practices, and affect coexistence and values. It also provides us with the tools to bring these concerns into a pro-active co-production method of project design with the community.

In addition to helping us conceptualize co-production for future projects, a territorial focus also emphasizes that understanding historical territorial transformations is essential to contextualize current local ecological knowledge, attitudes, and practices. Historical changes can shape current knowledge and practices, and may contribute to socio-ecological memory [28–31]. Socio-ecological memory can be held in the territory as much as in the mind or the collective culture, because these interact dynamically to produce ongoing knowledge and action [32]. The capacity to retain, remember, or reconstruct socio-ecological memory can be essential for ongoing adaptation [29,33]. Since Spanish colonization in the 1500s, central Chilean social structures and landscapes have been transformed through large-scale conversion to agricultural and silvopastoral land uses [34], the institution of latifundia systems consisting of a large private property (*fundo*) worked by mestizo *inquilinos* (tenants, similar to serfs) [35,36]. One of the results of these transformations was also the loss of native guanaco populations in most of central Chile [37,38], with only small populations surviving primarily in the high Andes [39,40]. Over the past hundred years, dramatic changes to land tenure and production systems included the Agrarian Reform, which expropriated latifundia and gave them to peasant cooperatives, the coup d'état which largely reversed these reforms post-1973, and the implementation of an extreme neoliberal economy which has promoted modern agribusiness for farmers with capital [35,41], micro-entrepreneurship for farmers without capital [42,43], and the progressive elimination of traditional livelihoods through a combination of economic incentives and conservation measures [36,44].

Traditional non-market extensive cattle production has been put under particular pressure both from an economic development perspective and from a conservation perspective. INDAP (the government office for agricultural development) has promoted the intensification of cattle production and encouraged its profit orientation, while CONAF (the government department in charge of protected areas) has circulated the idea that cattle are bad for woodlands and need to be removed from extensive pasture for conservation reasons [44,45]. From a desire to be pro-environment, many private landowners of *fundos* have moved to end pasturing rights preserved from the pre-Agricultural Reform era. In addition, a recent series of droughts and wildfires in central Chile, associated with climate change [46,47], have reduced available forage for livestock production. Thus, traditional

livestock farmers (*ganaderos, crianceros*) and *arrieros* (local people specialized in caring for livestock and transporting goods and, currently, tourists to the mountains) livelihoods and practices are under pressure from many directions, and their local ecological knowledge and socio-ecological memory risks being ignored and eventually lost.

The goal of the present study is to assess the territorial perceptions and practices of mestizo peasant cowboys (*arrieros*) and livestock farmers (*ganaderos*, *crianceros*) in the vicinity of a guanaco (*Lama guanicoe*) rewilding project being planned and implemented in a private conservation sanctuary in the central Chilean Andes. Our goal is to identify relevant local ecological knowledge and practices of cowboys and livestock farmers to identify productive angles for engagement in co-production approaches to implementing guanaco rewilding.

We make several predictions. First, since the number of livestock farmers and *arrieros* has been declining throughout central Chile, their age is increasing, and climate change has been putting pressure on traditional extensive cattle pasturing and transhumance; we therefore expect that territorial practices forming *arriero* local ecological knowledge will demonstrate evidence of recent historical losses, being highly uniform and not very diverse. A loss of knowledge and practice diversity could further point to difficulties with coping or adaptation. Second, because livestock farmers and *arrieros* and their traditional livelihoods are also under pressure from socio-economic changes and climate change, we predict that *arrieros'* and livestock farmers' perceptions of guanacos and their reintroduction will focus on potential conflict, and thus be negative. Similarly, we expect perceptions and practices related to extant native and introduced carnivores (pumas [*Puma concolor*], foxes [*Lycalopex culpaeus*], and dogs) to be negative and conflictual. However, since guanaco reintroduction could partially relieve the effects of carnivore predation on livestock, we will be attentive to this or other possible benefits that may be perceived to emerge in the interaction between different conflicts.

## 2. Methods

### 2.1. Study Site

The locality of Cajón del Maipo is part of the Commune of San José de Maipo, located in the Andean part of the Metropolitan Region of Chile (Figure 1). While it comprises a large area (almost 5000 km$^2$), most of it is uninhabited mountainous territory. The commune is administratively divided into 23 localities and has a total population of 18,189 [48]. The socioterritorial configuration of the 23 localities is highly associated to four main rivers that form the valley of "Cajón del Maipo": the Maipo river, Yeso river, El Volcán river and Colorado river. The commune is also characterized by having an important amount of ecotourism and one state conservation area, Monumento Natural El Morado, with 3009 hectares. The public conservation area is supported by two main private conservation areas: Santuario de la Naturaleza Cascada de las Ánimas, with 3600 hectares, and Santuario de la Naturaleza Lagunillas y el Quillayal with 13.426 hectares.

### 2.2. Mixed Methods

We use a mixed methods approach to, first, understand basic sociocultural and socioeconomic variables of the territory and, second, understand specific issues emerging from the semi-structured interviews through a questionnaire. The semi-structured interview has the advantage of qualitatively characterizing general patterns about the territorial practices and their relation to the rewilding project. The questionnaire (or structured interview) has the advantage of quantitatively exploring some of the variables evidenced in the semi-structured interviews, understanding specific patterns about the territorial practices and their relation to the rewilding project. In this way, the interviews informed some of the questions in the questionnaire, but the two methods focused on obtaining different kinds of information. The two methods were thus in part complementary and in part comparative [49,50].

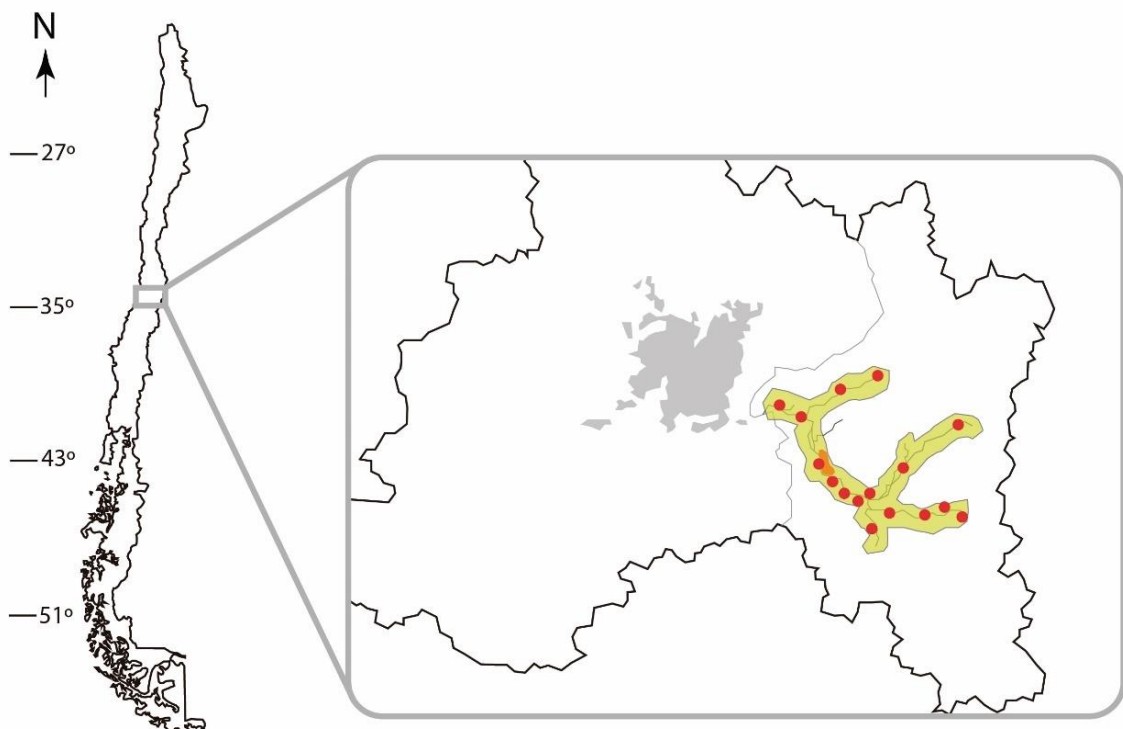

**Figure 1.** Study site conducted in Región Metropolitana, central Chile. Gray represents the city of Santiago de Chile, yellow represents the locality of Cajón del Maipo, orange represents the main town of the locality, San José de Maipo, and red dots represent the localities surveyed by semi-structured interviews and/or questionnaire.

### 2.3. Interviews

We first conducted semi-structured interviews through a set of questions with three main sections: the main territorial practices conducted by the interviewee, the adaptive capacity of people to a recent flooding event and the perception of people about a rewilding project that we are developing in the site.

We started with previous contacts in the Cajón de Maipo and then applied a snowball sample to identify key interviewees. We defined key interviewees as people with significant experience in any kind of territorial practices carried out in the site. Most of these people identified themselves as either *arriero*s or *ganaderos*, or as having formerly practiced those trades.

In total we interviewed 15 people from eight localities of Cajón del Maipo shown in Figure 1. Fourteen of them were male and one was a woman. Respondents were advised orally that their participation was anonymous, voluntary and that they could stop answering at any time. All respondents agreed to have their interviews recorded. The recordings were transcribed by hand.

### 2.4. Questionnaire

The information collected through the semi-structured interview was used to elaborate a structured questionnaire with three sections: section I asked about specific perceptions of people who conducted territorial practices in the study site about the rewilding project, section II focused on the main characteristics a rewilding project should have and section III focused on the management of dogs via responsible dog ownership because uncontrolled and feral dogs must be managed to allow guanaco reintroduction [51]. Questions in the last two sections are designed to assess the tacit models of conservation motivation held by the respondents [45,52,53]. There are four main tacit models corresponding to different assumptions about what motivates people to change behavior in conservation contexts,

including both personal motivations and responses to institutional settings [45]. Diagnosing the main tacit model held by a community may be useful to design culturally appropriate conservation interventions and programs [53].

Questionnaires were applied to 31 *arrieros* or livestock farmers in the Cajón de Maipo area through snowballing. Three of the respondents were previously interviewed for the semi-structured interviews but the other 28 were not included in the semi-structured interview. Respondents were advised in writing and orally that their participation was anonymous, voluntary and that they could stop answering at any time. The surveys were read aloud to respondents due to variable levels of literacy among the target population. Answers were entered manually into a Google form by the survey team for recording and visualization.

### 2.5. Data Analysis

We analyzed qualitative data through codifying the interviews with Atlas.Ti 8.4. Quantitative data from the questionnaire was analyzed through one-way ANOVA and Pearson's $X^2$ to see differences between groups of questions, using RStudio (R version 3.3.3 (2017-03-06)). To assess which tacit model is best supported by the results from sections II and III of the questionnaire, we cross-referenced support for each extreme of each axis of the schema of tacit models of conservation motivation, to correspond to the four described models. Three questions per each of the four models represented each extreme of the personal and institution axes (see Root-Bernstein, 2020, for more details about the type of questions for each model [45]); therefore, to create a simple index of support for each tacit model of conservation, we simply multiplied them together. During data analysis we decided that one of the questions about rule-based behavior seemed to specifically support the Normative Model but not the Uniformity Model, so we counted it only in that category.

### 2.6. Ethics

The research was led by MR-B who is based at the CNRS in France, where researchers are bound by a personal deontological responsibility. The survey met the standard of CNIL (2018) [54], the National Commission for Information Technology and Civil Liberties, and therefore did not require ethical approval.

## 3. Results

### 3.1. Interviews

#### 3.1.1. Local Territorial Practices in the Mountains

In the past, *arrieros* carried out a variety of practices that arose as adaptations to the remote mountain conditions:

> " … 200 [years] the peasants have been here and those peasants, over time, became arrieros. What does that mean? That they started to herd livestock, that they were the only ones who had the capacity and the shrewdness, that they knew the strategic points where to pass through and where to go ( … ) Arrieros transported and bartered, in the old days, barter, here and there and thanks to this, the conversion [to being arrieros] started."

The main characteristic of *arrieros* is that they know the territory intimately but do not own any land of their own:

> "*Arrieros* in general herd their animals in lands that are national lands or some *fundo* [private landholding]. If an arriero had a *fundo*, he wouldn't be an arriero, let's start with that. An arriero is a person who has certain territorial characteristics and lives around moving animals in the mountains. They might be cattle, sheep, goats, or even horses."

*Arrieros*, whether they herded their own livestock or worked for a *fundo* [large landholding] caring for the owner's livestock, often spent long periods alone in the mountains and thus had to master many skills:

"I spent the whole month in charge of the animals, every day I was watching them, some pregnant cows would appear, I enjoyed myself, I had to find the little calves, lots of things, milk the cows to get rid of the milk because sometimes they accumulate a lot of milk ( . . . ) Sometimes you miss friends to go around with you ( . . . ) You are almost always by yourself, the fundos [landholdings] are closed, sometimes in the evening you have to herd the livestock and close them up in the corral and lock them in and then . . . sleep alone, the next day make a fire, put on the kettle, make your coffee, as was one's tradition, in the countryside, the countryside . . . Yes men of the countryside, many can . . . I for example . . . had to do everything, let's say cook for myself, make bread, so many things, lassos, the equipment for the horses, the reins . . . the sheep skins for the saddle."

We found that the diversity as well as the intensity of territorial practices showed evidence of decline, with many people referring to arriero practices that they no longer carried out. The current practices consisted mainly of keeping goats, horses, cattle and sheep and providing touristic services for people who want to, for example, ride horses to specific places in the mountains. The main reason for this loss of the range of arriero practices and skills is the disappearance of the people who know how to farm livestock. Now, they are old and the new generations are dedicated to other activities.

"They are disappearing, the people who know the mountain passes, who know how to recognize where to take refuge, who know how to read the nature of a place. So these kinds of people are disappearing. As they disappear, history is lost."

"Yes, it has changed . . . of course, the difference is that for example before, the older people took their children out very little so that they would learn, so from one moment to the other livestock raising was lost, why? Because of course if . . . I bring my son [to the mountains] because they learn here, they like it, but imagine if I had never brought them . . . "

This lack of intergenerational transmission, according to their accounts, is related to the current drought that central Chile is suffering, which pushes livestock farmers to other kinds of activities outside the mountain. The drought makes it more difficult to raise livestock and less attractive in economic terms. One arriero said:

"Here there was the El Sauce stream, but it grew a lot [the stream], you cannot go on walking, you had to go through the trees, it rained, it snowed a lot and now we are scared because we are already drying up."

In addition, one interviewee mentioned a political climate that was not in favor of smallholder livestock raising:

" . . . the government wants to exterminate the livestock farmers."

Most *arrieros* also made a living in other ways, including work related to mining companies, due to the presence of two gypsum mines, work related to transport of products between Argentina and Chile and the movement of specific groups interested in visiting some places in the mountains, such as mountain climbers and geologists. Now several such practices have also almost disappeared because they are less developed or because the mining companies have modernized their equipment.

### 3.1.2. Local Adaptations to the Environment

As suggested above, being an *arriero* was itself a historical adaptation to living in remote mountains. *Arrieros* also displayed many more recent adaptive responses related to coping with changing circumstances, although the forms they took were mixed. Most *arrieros* reported changing their economic activities, a form of coping with a changing socioeconomic and ecological scenario. The main environmental driver for adaptation is the reporting of an increase in puma attacks, as discussed below. Interviewees perceived

that raising livestock is an unviable economic activity and can even be pointless, since most of the offspring or even adults were predated, leading them to sell their livestock and give up livestock raising.

While the increase in pumas can be considered as one of the main drivers, livestock farmers mentioned other factors such as the conversion of large landholdings from traditional extensive pasturing areas to touristic and conservation sites. The immediate consequence of less pasturing area is a decrease in the number of livestock, which reduces the income that can result from livestock farming activity. We do not know the specific conservation and management measures of each of the several landowners in the study site. In Cajón de Maipo there are two formal private conservation areas, one of them recognized as being for livestock pasturing but which decided, two years ago, to decrease the number of livestock from local livestock farmers and, in the near future, eliminate livestock pasturing completely. Other areas were now off-limits due to extractive industrial development:

> "Now all those passes are closed and prohibited. Here it is prohibited, not just anyone can enter because of the issue of the gas pipeline."

These problems were part of a longer history of changes to the ability to access pasture:

> "I had [goats] for ten years . . . [until] Pinochet [dictator from 1973 to 1990] stopped us. The frontier was closed for three years and so the mountains were closed for the spring pastures."

The other main driver of adaptation that livestock farmers mentioned is the drought that central Chile is suffering. The water stress on many pasture lands was observed by several livestock farmers and *arrieros*. The consequence of the drought is not only for livestock, but also affects some livestock farmers over normal water consumption. The decrease in the productivity of pasture lands ultimately obliges the livestock farmers to also decrease the number of livestock.

One of the main activities to which some livestock farmers and *arrieros* are converting is the tourism sector, as a way to value their knowledge of the mountain territory:

> "There was an opportunity for reconversion to the extent that tourism was a source of opportunities that could be added on to the lifestyle of the *arriero*, which indeed involves a lot of sacrifice, because it is a very dry and hard mountain range. So, beyond the issue of grass, water and the mines, this makes the [activities] of an *arriero* increasingly hard to carry out."

In terms of coping with natural disasters such as a serious flood that occurred in 2021, some *arrieros* reported that the community did not come together to help one another, while others reported that they both gave and received help from the community. As one interviewee stated:

> "In some areas yes they are united, for floods and things like that they are united, but the unity lasts for a while and then they come apart. There are sectors that are more united, [in] Río Colorado they have made associations and things like that . . . In San Gabriel it is not so united and in San José just from time to time."

### 3.1.3. Local Ecological Knowledge

Interviewees agreed that the environment and climate had changed significantly in the past decades. They nearly all pointed to a distinct decrease in precipitation, and some also describe how quickly high Andean springs are drying up:

> " . . . in the past it was very very rainy, it rained day and night, two days, three days, there were huge snowfalls . . . and the old people and housewives had to make a path with a shovel to get to the houses in the *fundo* [landholding] . . . "

> "The drought is the biggest threat. It will be just like in the north, the desert. Maybe I won't see it, but my children will. I remember that 20–30 years ago there was like 40 cm of snow and now imagine, it has all dried up. Now you go to the

mountain range and it is just like being here, before wherever you went in any ravine it gave you pleasure to wash your face and drink water, but now you have to go for kilometers to find a ravine with water, it has all dried up."

*Arrieros* also had a knowledge of weather and mountain conditions and terrain that was developed through experience:

"Well, what happens is that the *arriero* has to have an education. But an education, you see it as . . . they laugh because what education is one going to have in the mountains, but I stupidly call it that anyway. He has to have an education, let's call it that or we can change it and say, the respect that you have to practice in the mountains, a special respect. The *arriero* has to know it ( . . . ) [I]f it's really good for going on a trip, it's because there are some tourists who are going to pay me like 40,000 pesos per person and I have to spend like two days up there and it's good weather, but if you see that it is bad [weather], better not to get mixed up in it."

There is also a recognition of specific community and ecosystem dynamics concerning the rainfall and snow. The drought has decreased the water supply of springs, which normally have water year-round. Some *arrieros* and livestock farmers recognize that when insufficient snow falls, the springs are not recharged, and some community-level changes occur. For example, one *arriero* said:

"When there is more snow, the springs increase, they flowed to the Maipo river and through it, the river branches that you can find around there . . . they had more water [ . . . ] and then, it snows a lot up there, it rains, it snows, the tagua [possibly Fulica armillata] will arrive here downstream, through the ravines and they are going to go to the Maipo river and the river will grow downstream."

Concerning the knowledge contained by local communities, and particularly livestock farmers and *arrieros* about the guanacos, there are differences in the specific properties about the species, such as distribution, number of individuals in the past, preferred habitats, among others. Some interviewees said there have been no guanacos in the specific place where they lived for at least the past 60 years, a fact that, for them, confirms their hypothesis about the inability of guanacos to live close to towns. Others, located in places closer to the frontier between Argentine and Chile said they saw guanacos five to six years ago, although they referred to them as isolated individuals who rapidly returned to Argentinian territory. There is a more consensual notion about the preferred habitat of guanacos, claiming that they avoid anthropogenic landscapes and prefer high mountains or vast open areas such as the ones found in Argentina, but which are scarce in central Chile.

" . . . the guanaco snatched himself off to the mountains because they belong to the high peaks."

" . . . [the guanacos] would be up in the mountains so they wouldn't influence me at all."

Some of the livestock farmers also recognize the trophic relation between the puma and the guanaco, which is part of the advantages a few of them declared if there is a reintroduction of the species.

"For me it would be good that there should be guanacos, for all the livestock farmers, because they are going to provide meat to the pumas. We will save ourselves because [the puma] hunts mainly guanaco."

"[Guanacos] would just be meat for [pumas] ( . . . ) That would be good because the pumas wouldn't attack our colts and would have their own food, obviously they wouldn't come down here."

" . . . it is to be hoped that there will be more variety of hunting for the puma, and the other is that equally nature is prettier like that with animals that are different,

different from whatever the horse or the cow that we are accustomed to but those animals, for instance a fox or a couple of guanacos would look nice."

There is local knowledge concerning the behavior of other more conspicuous species, such as the condor (*Vultur gryphus*) and puma. The condor is a scavenger recognized as being constrained in central Chile because of the scarcity of meat predated by the puma. Some livestock farmers and *arrieros* recognized that this food constraint generates a behavior in which the condor attacks recently born calves, decreasing the survival rate of calves each year. This behavior is described as increasing over time.

> "The condor, which around here they call a vulture, was killing the offspring right away, so the cows were giving birth and before they even gave birth the condor picked them off."

> "Many people say that the condor is a scavenger, and yes, it is a scavenger, but an animal doesn't die every day to maintain so many condors, so the condor has to hunt, it's the law of nature ( . . . ) They kill our calves, the cows give birth and there are no more calves."

Pumas have shown similar changes in behavior and prey selection, although there were differing accounts as to why. Pumas specialized in recently born foals, but had also started to eat calves. Respondents highlight that the pumas almost never attack cattle and when they begin to do it, it represents a high risk for humans because they are starving and they can, then, attack people.

> "[The pumas] eat foals and calves (the latter they didn't eat before, because they say that the puma prefers prey that runs, and the calves are too curious [to run away])."

> "Before, pumas didn't eat calves. What happened is that they are these tame pumas, the ones they [supposedly CONAF] release and bring from down below [in the lowlands]. Once the puma starts to eat calves, it starts to move among people. It will start to eat people next. Before, there were only mountain pumas that when they saw you with dogs they would run away."

It is important to note that the interviews were made in a wide territory, comprising more than 80 km from one end to the other, so ecological differences can be found according to the place where the people interviewed have lived.

### 3.1.4. Perception of Guanaco Reintroductions

Perceptions of potential guanaco reintroduction were generally fatalistic rather than negative, with only a minority expressing positive views of the proposal. In other words, most *arrieros* were not against the idea, but they were convinced that it would not work. Several reasons were given for it not working, which we can analyze from sociocultural and ecological perspectives. Concerning sociocultural reasons, the main idea was that the sole presence of guanacos will stimulate the return of historical hunting activities by mountain livestock farmers and *arrieros*. One livestock farmer said that:

> "Many people will come and they are going to bring them and all that, like the people who rustle livestock [ . . . ] as they used to call them in the old days, they are going to come and hunt, they are going to go around at night."

> "Down here they are going to eat them or they are going to scare them. The dogs themselves will eat them."

In addition, many *arrieros* and livestock farmers said that the guanaco is not part of the local ecosystem. They argued that human presence and the small amount of wide-open mountain grasslands impede the presence and reproduction of guanacos. As a consequence, there is an idea that any project of reintroduction will end in guanaco herds moving to the Argentinian frontier, escaping from the small and crowded places of the central Chilean mountains. One *arriero* said:

"It is not possible, I believe that you cannot do it, guanacos move away from people, they go away, they stay away, the guanaco is an animal that loves to live in solitude, in their mountains, for example in Argentina."

There were other reasons for opposing the project. One was that the presence of guanacos would stimulate land protection, further excluding livestock and closing the historical passes for livestock farmers and *arrieros*, jeopardizing the traditional activities of the territory.

"This idea of putting [guanacos] in a site is a business for the owners. ( . . . ) Imagine, to give you an example, here in Laguna Negra they put in some guanacos, the chamber of tourism takes charge because they are really interested in this area, they install gates, the *arrieros* won't be able to enter there, nobody will be able to enter, only people in vehicles up to a certain area, they will charge an entrance fee and all that ( . . . ) What do we get out of it? Absolutely nothing, we will have to take out the livestock and will be left in a worse state than we are now."

Another socioecological reason for opposing the project was the idea that guanacos will compete with livestock for pasture. Pastures are thought to be in danger because of the drought that central Chile has been suffering, affecting the production of vegetation in the mountains. The reintroduction of a herbivore is seen as an additional component that will further stress the grazing lands, reducing the amount of pasture for livestock. One *arriero* said that:

"The owners of fields would be negatively affected. Suppose that 100 guanacos arrive in a field [ . . . ] they will eat all the pasture and the owner will not be able to rent it [as livestock pasture]."

### 3.1.5. Perception of Carnivores

Perceptions of relations with carnivores pointed to an increasing and persistent conflict. We did not obtain any information about relations with native foxes (*Lycalopex spp.*). Dogs were not frequently mentioned, although there were references to the relationship between dogs and livestock, which was extended to guanacos, where it was expected to have a detrimental effect. One livestock farmer believed that feral dogs will predate guanacos, decreasing the populations:

"Dog packs or the very dogs from the tourists themselves will end up attacking guanacos. There must be education, it is like a cycle. It is not only [enough] to arrive and reintroduce them. I think there must be education for livestock farmers and *arrieros*, and [you must] educate the community."

Pumas, however, are the focus of an intense conflict between carnivores and livestock farmers, as they were frequently mentioned for causing a problem by eating a large number of livestock. The livestock farmers mentioned the conflict with pumas as one that has increased during recent years. The main explanation for this, they almost unanimously claimed, is the reintroduction of pumas by the Livestock and Agricultural Service (SAG by its acronym in Spanish), the governmental institution that is responsible for agricultural and wildlife regulations. As a consequence of more frequent encounters with pumas, they said they lose a great number of livestock, mainly horses. One livestock farmer said:

"Raising horses has no benefit now because you have to wait one year for the mare to give birth and have the colts at home. When they are 9 months old, they are released and the next day a puma has eaten the offspring. [Raising horses] is plagued by pumas."

Some *arrieros* spontaneously suggested or admitted when asked about the topic, that guanaco reintroduction would be good for livestock raising because it would provide an alternate prey for pumas, reducing the conflict. While there is no reference in terms of

the historical presence of guanacos because of the historical local extirpation of guanaco populations, some *arrieros* and livestock farmers are conscious of the trophic relation between guanacos and pumas. One livestock farmer said:

> "[The reintroduction of guanacos] would help with puma [attacks on livestock], because the puma maybe will follow the guanaco, once the guanaco has arrived, it is going to prefer guanaco meat to foal meat."

Others pointed to other benefits of a less practical or tangible nature, such as aesthetic or contemplation benefits for the mountains and the people who live there. Another less mentioned benefit was the additional economic income as a consequence of the presence of guanacos. Some *arrieros* maintain, as part of their livelihoods, the touristic activities of horseback riding to the mountains. The presence of guanacos may benefit them in terms of opportunities for wildlife observation for the people that buy those services.

### 3.2. Questionnaire

#### 3.2.1. Socio-Economic Profile

Some 31 individuals from diverse locations within the territory answered the questionnaire, of whom 27 (87.1%) were male, aged between 20 and 85. The modal age, also the most represented, was 57. The range of time that respondents had lived in the locality where they were interviewed in the Cajón de Maipo was between 1–60 years, with the mode being 30 years. Sixteen respondents (51.6%) identified themselves as *arrieros*, 29 (93.5%) as *ganaderos*, 10 (32.3%) as tourist operators, five (16.1%) as farmers, and 10 (32.3%) as a variety of other jobs. These choices were not mutually exclusive because many people combined various activities. Almost all respondents reported living in a family home with other inhabitants in the same home, with between 1–12 residents, with a mode of 3. Only five (16.1%) respondents had finished high school (the highest level of education recorded), with the majority (13 people, 41.9%) having partially completed elementary school. The majority of respondents (27 people, 87.1%) reported that the principal source of heating in their home was firewood. Twenty-one people (67.7%) reported not having internet in their homes.

#### 3.2.2. Plant and Animal Resources

The most common reported regular uses of local natural products included firewood (14 people, 45.2%), and medicinal herbs (15 people, 48.4%). Six people (19.4%) reported using no forest products. A majority of respondents also reported growing vegetables for their own consumption, many cultivating potatoes, maize, tomatoes and pumpkins, among other produce. Fifteen people (51.6%) reported owning no cattle. For the other respondents, reported numbers of cattle owned ranged between one and 34, with the mode at 6.5 cattle. For goats, eight people (25.8%) had none, with the number of goats reported owned by the rest of the respondents ranging from 10 to 380, with a mode of 100. The number of horses owned ranged from 0 to 40, with a mode of 7. Other animals that were reported by smaller numbers of respondents included chickens, ducks, geese, sheep, mules, and pigs. The number of dogs (used for different purposes but not including animal herding or protection) owned ranged from 0 to 20, with a mode of 4.

#### 3.2.3. Knowledge of Guanacos

When asked if they had seen guanacos in the area recently, 21 (67.7%) people said "no" and 10 (32.3%) said "yes". When asked if they had seen guanacos in the past in the area, this was reversed with two-thirds of respondents saying "yes" and one-third saying "no". When asked when they had last seen guanacos, answers ranged from one month ago to 45 years ago. In response to a question about where guanacos live, the only option for which all responses were either "agree" or "strongly agree" was "the high Andes (where it snows)"; however, only the response to whether guanacos live in woodlands was significantly different from the other answers, with a tendency towards negative ("disagree") responses (ANOVA, Table 1).

**Table 1.** ANOVA compared to the response to whether guanacos live in woodlands. Significant *p*-values are shown in bold (indicating significant difference to the "woodlands" response).

|  | Df | Sum Squares | Mean of Squares | F Value | *p*-Value |
|---|---|---|---|---|---|
| Grasslands | 1 | 4.432 | 4.432 | 3.426 | 0.0765 |
| Shrub habitats | 1 | 11.699 | 11.699 | 11.699 | **0.0061** |
| *Espinals* | 1 | 6.393 | 6.393 | 4.942 | **0.0359** |
| High mountains | 1 | 7.963 | 7.963 | 6.155 | **0.0205** |
| Low mountains | 1 | 7.681 | 7.681 | 5.937 | **0.0226** |
| Central Chile | 1 | 0.329 | 0.329 | 0.254 | 0.6189 |
| Residuals | 24 | 31.051 | 1.294 |  |  |

### 3.2.4. Attitudes to Guanaco Reintroduction

Three-quarters (~75%) of respondents supported the idea of guanaco reintroduction. A majority (14 people, 45.2%) reported that they "agree" that reintroducing guanacos in the Cajón de Maipo would be a good idea, with the next largest response being "strongly agree" (Figure 2). When asked more specifically if guanacos should be reintroduced into the espinal habitat, a savanna-like vegetation formation dominated by *Acacia caven*, answers were significantly different (t-test, t = 22.23, df = 61, *p*-value < $2.2 \times 10^{-16}$) and more nuanced, with a total of 57.2% saying "agree" or "strongly agree", revealing more "I don't know" (increase of three people) and negative responses (increase of two people).

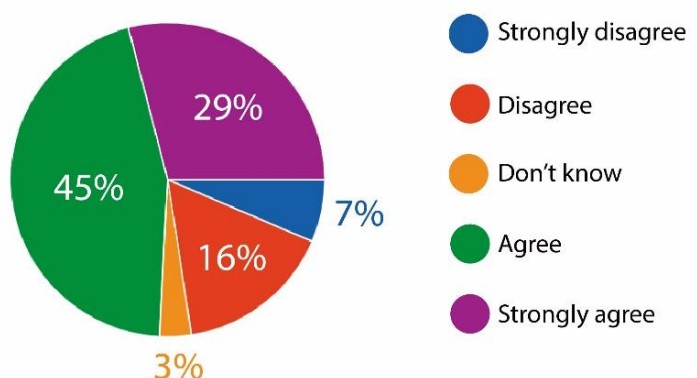

**Figure 2.** Responses to the questionnaire question "Do you think that introducing guanacos in central Chile is a good idea?" All 31 participants answered.

When asked about reasons for supporting or not supporting guanaco reintroduction, most reasons were supported. A majority thought that guanaco reintroduction was compatible with the identity of the Cajón de Maipo territory (Pearson's $X^2$ test, $X^2$ = 8.7935, df = 2, *p*-value = 0.01232). Although a majority thought that guanacos should not be reintroduced in an area with human presence (such as the Cajón de Maipo) (Pearson's $X^2$ = 18.926, df = 2, *p*-value = $7.767 \times 10^{-5}$), at the same time, most respondents agreed with the statement that the reintroduction "would be good because [the guanacos] disappeared due to humans" from what was in fact their former natural habitat (Pearson's $X^2$ = 17.556, df = 2, *p*-value = 0.0001541). Most respondents agreed that guanaco reintroduction would make the landscape more beautiful (Pearson's $X^2$ = 9.8785, df = 2, *p*-value = 0.00716), and that it was good as long as it brought economic benefit (Pearson's $X^2$ = 6.5101, df = 2, *p*-value = 0.03858), or benefit to the local animals and plants (Pearson's $X^2$ = 7.3746, df = 2, *p*-value = 0.02504).

When asked to consider possible outcomes of guanaco reintroduction, respondents' support for reintroduction varied somewhat but not dramatically. When asked to consider possible economic benefits of guanaco reintroduction from tourism or exploitation of their fibre (wool), 71% of respondents were positive about a reintroduction (10, 32.3% "agree"; 12 people, 38.7% "strongly agree"). When asked to consider possible cultural benefits related to increased visibility for *arriero* activities focused on guanacos, a total of 64.6% of respondents were positive about a reintroduction (six people, 19.4% "agree"; 14, 45.2% "strongly agree"). When asked about possible changes in daily life because of guanacos living in proximity, 54.9% continued to be positive about a potential reintroduction (11 people, 35.5% "agree"; six, 19.4% "strongly agree"). Across these three questions, the proportion of negative responses was fixed at nine people (although notably these were not the same people for each question), with an increasing share of uncertainty ("I don't know" increased from zero to five people). The responses to how guanacos might change respondents' daily life differed significantly from the responses to potential economic benefits of guanacos (ANOVA, Table 2).

**Table 2.** ANOVA comparing attitudes to guanaco reintroduction under three scenarios of possible outcomes. Here, the variable "economic outcomes" is compared to "cultural outcomes" and "daily life outcomes" and is significantly different from responses to "daily life outcomes" (which had more "Don't know" responses). The significant p-value is shown in bold.

|  | Df | Sum of Squares | Mean of Squares | F Value | *p*-Value |
|---|---|---|---|---|---|
| Cultural outcomes | 1 | 2.50 | 2.496 | 1.592 | 0.217 |
| Daily life outcomes | 1 | 9.55 | 9.549 | 6.092 | **0.020** |
| Residuals | 28 | 43.89 | 1.568 |  |  |

### 3.2.5. Design of a Reintroduction Project

All answers to questions regarding the tacit models of conservation as applied to a hypothetical guanaco reintroduction project in Cajón de Maipo had a very high rate of agreement, after inverting the questions phrased in the negative, which makes it essentially impossible to assess differential support for one tacit model of conservation or another. We thus did not look at these results in any detail. In more specific questions about this particular project, almost all respondents were against puma reintroduction (28 people, 90.3%); around half (16 people, 51.6%) preferred to work with government entities other than SAG, the agricultural and wildlife authority; and a small majority (20 people, 64.5%) did not want to see large landowners receive direct benefits from the project.

### 3.2.6. Responsible Dog Ownership

These questions were also designed along the lines of the tacit models of conservation, to understand how respondents viewed their motivations and those of their peers in terms of inducing behavioral change related to responsible dog ownership. Summarizing the index of support for each tacit model shown in Figure 3, we find the highest support for the Implication Model at the intersection of discretionary action and context-dependent action.

**Figure 3.** Index of support for each tacit model of conservation motivation. The Xs score which end of the axis is supported by the majority answer for each question. The numbers in the boxes show the multiplication of the scores. During data analysis we decided that one of the questions about rule-based behavior seemed to specifically support the Normative Model but not the Uniformity Model; we therefore count it only in that category.

## 4. Discussion

This study advances our understanding of how guanaco reintroduction projects in central Chile may impact local people, especially those who make a living from the territory, and whose local ecological knowledge is crucial for designing a successful conservation and reintroduction project.

*Arrieros* and livestock farmers are living in a changing socio-ecological context. Our first prediction was that we would observe a reduced diversity of territorial practices among our sample. We found that a wide variety of skills, practices and knowledge were traditionally associated with being an *arriero*, from local ecological knowledge, knowledge of terrain and weather, knowledge of animal management and care, to a wide variety of manual skills and crafts. Most interviewees reported having given up several of their practices, or having reduced the number of livestock they raised, or anticipated abandoning their practices. Many others had reconverted their skills as tour guides or changed work altogether. They also reported that the younger generation was not learning *arriero* practices. At the same time, as revealed by the questionnaire, the remaining practices were not extremely uniform, as they included raising a range of animals, collecting a wide variety of wild plants, and cultivating several other plants for subsistence, in addition to wage labor in a variety of other jobs. Their farming and income-generating strategies were thus diversified, similarly to central Chilean peasants studied in other localities [44].

Our second prediction was that this loss of territorial practices would contribute to difficulties in coping with and adaptation to socioecological change. While the interviewees described links between loss of practices and the challenges of coping and adaptation, the direction of causality is not completely clear. The immediate cause of loss of territorial practices is the loss of transmission opportunities as people move into other forms of work. The drivers of these changes in livelihood are in turn related to climate change and increasing predation pressure from pumas. On the one hand, loss of transmission and progressive abandonment of *arriero* livelihoods could be seen as a failure of adaptation to changing environmental and socio-economic conditions. On the other hand, it may be the loss of transmission itself, and the resulting loss of community knowledge, that leads to this failure of adaptation. The process is likely to be circular and self-reinforcing. A different perspective, however, could be that changing livelihoods, moving towards tourism and entirely unrelated jobs such as truck driving or working in mines, do represent a form of effective coping and adaptation, since they permit people to make a living

without migrating. We also found that the local community had a mixed or weak record of communal action to cope with local problems, but that did have the capacity to come together during crises.

Third, we predicted that pressures from climate change and socio-economic changes put pressure on *arrieros* and would thus lead them to see a potential guanaco reintroduction project in terms of conflict, and thus negatively. Some interviewees very clearly made this point, such as the individual who predicted that guanaco observing would become a private touristic business leading to further *fundo* enclosures, concluding "*What do we get out of it? Absolutely nothing, we will have to take out the livestock and will be left in a worse state than we are now*". Although reflecting concerns and trends in dispossession and loss of traditional access and resource rights across rural Chile (e.g., [44,55]), this negative and conflictual view was the exception.

It is interesting to note that questionnaire respondents' answers are comparable to similar questions about potential guanaco reintroduction projects posed to an educated urban elite in a previous study [11]. For example, in both studies most respondents agreed in principle with guanaco reintroduction, were somewhat unsure about the full range of habitats guanacos could live in but broadly supported reintroduction into an *espinal* habitat, and supported the reintroduction on aesthetic, economic, and moral grounds. Although positive views were the majority, a notable minority had negative views as suggested above. Further, the reduced sample size, when put into the context of a reduced amount of *ganaderos* and *arrieros* in Cajón del Maipo, takes on more relevance since these groups of people may represent the last exponents of the activity. Many *ganaderos* actually participate in local livestock organizations but with a reduced number of animals, showing that, for many of them, it represents more an inherited cultural activity than an economic one.

However, of particular interest was the fact that many of the interviewees simply thought that the guanaco reintroduction project would not work. This was for two reasons. On the one hand, they pointed to a conflictual reason: the guanacos would be (illegally) hunted, or eaten by dogs. These of course are real concerns, which should be resolved before any reintroduction can take place [51]. On the other hand, an even more common reason was that their understanding of guanaco behavior and habitat preference was that guanacos would not stay in the Cajón de Maipo, which had too many human settlements and activities, and would move far up into the Andes or cross into Argentina. It is, of course, difficult to assess whether these observations and opinions about guanaco behavior represent real guanaco habitat preferences or reflect a shifting baseline effect [56] due to the absolute rarity of guanacos in the region. Our own observations, based on a trial reintroduction, suggest that guanacos establish home ranges in woodlands and shrubby lowland or low mountain habitats [56], while historical accounts suggest that guanacos used to migrate into the central valley lowlands every year [38]. In summary, despite some focus on conflicts that may arise between guanacos and hunters and dogs (represented as conflicts with other people, not with *arrieros* or livestock farmers themselves), a majority of interviewees and questionnaire respondents nevertheless thought that the reintroduction was desirable for a variety of reasons.

Fourth, we predicted that perceptions of pumas, dogs, and foxes would also be negative and conflictual. Interestingly, we heard little about dogs, and almost no mention of foxes, but rather condors were unexpectedly mentioned several times as problematic predators. Pumas stood out as representing a clear and salient conflict for *arrieros* and livestock farmers. Pumas' behavior was observed to have changed, with increasing predation rates on foals and the addition to their diet of calves. This was often explained in two ways: there were more pumas, and the pumas were "tame" and unafraid of humans. While most people repeated the common rumor that SAG translocates pumas by helicopter or hidden in official cars of the governmental agency into rural areas, we suggest that another explanation, also suggested by some interviewees, may be more likely. As the drought and other changes described above have reduced the number of livestock in the mountains–and

with the historical extirpation of guanacos– there are not more pumas, but rather fewer prey per puma. The pumas are hungry and have learned to overcome their fear of humans.

Finally, we predicted that *arrieros* and livestock farmers may see some benefits to guanaco reintroduction, for example in the possibility that they would provide alternative prey to pumas. This possibility was indeed spontaneously perceived by several interviewees, who saw it as a potential solution to the conflict with pumas. In the survey, a majority of respondents supported the project when informed about possible economic and cultural benefits to *arrieros*. Indeed, it is not really correct to say that they perceived a "silver lining" since their overall perceptions of the proposal were already that it was a nice idea–just that it would not work.

As always when dealing with local ecological knowledge and perceptions, we may notice that some explanations or observations conflict with what we think of as "objective" knowledge, and we have to consider how to contextualize the interpretation of this information. For example, to our knowledge SAG does not have a policy or a practice of translocating pumas into the mountains at night by helicopter, which we consider to be merely a widespread rumor or myth. It does, however, point to several things: a conflictual and distrusting relationship with SAG (also reflected in at least half of questionnaire respondents preferring to work with some other government partner), the sense that rural livelihoods are sacrificed to other values held by the elite such as nature conservation, and the observations that there are too many pumas for the available prey and that they have lost their fear of humans and changed their predation habits. We can also ask, as we referred to above, whether guanacos will really flee the Cajón de Maipo or whether their absence and rarity are an artifact of their historical extirpation, or how different processes such as habitat imprinting by guanacos may alter a current preference for avoiding human settlements. Our observations suggest that guanacos can prefer to live in wooded lowlands [56], which points to an opportunity to explore together with *arrieros* how guanacos may adapt to existing and changing territorial practices. It is not only humans who change their territorial practices in response to dynamic situations– large herbivores do so too (e.g., [57]), and the study participants are aware that condors and pumas also change their habits. Finally, by contrast, what may seem like a conspiracy theory that the government wants to eradicate smallholder cattle raising, mentioned by one interviewee, has been confirmed to us by CONAF, the Forestry Corporation in charge of environmental policy and protected areas, as government policy as of 2019 [44,45].

One of the weaknesses of the study is that the questionnaire respondents, as well as the interviewees, did not always state clear and decisive opinions or positions, but rather at different moments appeared to hold opposite opinions or support all possible justifications of all possible positions. On the one hand, this may represent a confirmation bias in which respondents and interviewees alike seek to avoid expressing a disagreement with the question, or say whatever they think the interviewer wants to hear. This does not entirely prevent them from expressing their own view, should it be contrary, but they do so in a round-about way by tentatively expressing a full range of views. While we tried to avoid leading questions and inverted some of the questions in the questionnaire, it is not clear that we were consistently able to avoid what may be considered a local cultural phenomenon of avoiding open disagreement and nuancing one's opinion to match the context. At the same time, we should distinguish fatalistic attitudes ("it will never work") from being against a proposal due to one's interests or values.

Taking this into account, responses to how the guanaco reintroduction project should be designed were particularly unhelpful as there was a notable agreement with every question. Without clear disagreement on some statements, it is impossible to determine which tacit model of conservation motivation is most supported. We thus chose to ignore this section of the questionnaire as it appeared to have been a methodological failure. By contrast, the perhaps more salient and less speculative issue of how to motivate change in responsible dog ownership–which also may have been perceived as an issue not at the core of our interests as interviewers, and thus subject to less confirmation bias–did elicit a clear

range of different answers to different questions. In that section, we see a preference for approaches that recognize the specific territorial, cultural and socio-economic context of *arrieros* and livestock farmers, and that acknowledge that people behave in a discretionary manner, doing whatever they want in spite of laws and regulations. This combination of personal motivation and response to institutional (regulatory) environments is equivalent to the "Involvement Model". In this tacit model of conservation motivation, behavioral change is most likely to be accepted or to be successful if it involves local stakeholders in creating the conservation contexts in which they will choose how to act. This can include variations in co-productive and participative approaches (e.g., [58,59]). In other work in central Chile [52], we found that peasant farmers may support different approaches for motivating behavioral change on divisive vs. consensual topics. The Involvement Model was supported in that study by male smallholders with a traditional lifestyle and little community experience with participative conservation processes, for consensual or common-sense issues. This may also be a good description of our sample in this study in response to the well-established common-sense topic of dog management, while leaving as an open question whether they would also, as in the previous study, be more likely to support a values-based model (Persuasion or Normative) for the more open-to-debate issue of guanaco reintroduction. Further research would be needed to establish if the profiles of adherence to the various tacit models of conservation in [52] also hold true in this region.

Finally, to return to the framing of territoriality, we find that although *arrieros* may be coping with and adapting to socio-ecological change by abandoning traditional territorial practices, this presents an ironic and negative consequence of livelihood adaptation, in which socio-ecological territorial practices and knowledge are lost and replaced by a different kind of territorial knowledge based on modern infrastructures (e.g., truck driving, working in mines). This leads to a potential loss of cultural heritage, valuable local ecological knowledge and social-ecological memory, and equally could have unknown repercussions for the territory itself, in terms of its ecological processes and structures. The various pressures leading to reduction in herd sizes and pasture areas may have unexpected context-specific negative ecological consequences, including the collapse of predator populations, simplification of trophic webs, slowing of nutrient cycling due to loss of herbivory, and so on, especially in the current absence of guanacos (e.g., [60,61]). In the absence of government support for the territorial practices of *arrieros* and livestock farmers, we believe it is important for private conservation projects such as the one considered here to incorporate participatory and co-productive methods to restore both ecological and social components of territories–that is, both ecological processes and meaningful places. However, the current conservation programs are reducing the number of livestock in a system where the *fundo* owners rarely consult the workers and livestock farmers that rent some of the grazing pastures. Future studies must understand and identify what kind of territorialities are deployed by private conservation areas and other landowners who are interested in developing conservation actions and how they are related to other territorial practices identified by the present work. If the actions are poorly co-produced, a conflict is likely to arise with direct and negative consequences for the success of these actions.

As highlighted by many of the study participants, guanaco reintroduction can provide meaning to a changing landscape through the restoration of aesthetic, moral and economic values. We believe that these go together, since it is only territories that are used, and in which humans and other species interact dynamically and meaningfully, that provide us with the memory base and knowledge to conserve, sustainably manage, and adapt to a changing environment. There is an opportunity to put into practice some of the social-ecological memory shown by many *arrieros* and livestock farmers to re-establish positive ecological links, from the conservation perspective, between livestock management, guanacos and the broader ecosystem.

**Author Contributions:** M.G.-G. and M.R.-B. developed the research, M.G.-G., B.S.R., J.F. and A.E. conducted the fieldwork, M.G.-G., M.R.-B., T.E.R. and B.S.R. analyzed data and M.G.-G., M.R.-B. wrote the manuscript. All authors have read and agreed to the published version of the manuscript.

**Funding:** This research was funded by Grant ANID/BASAL FB210006 and the Global Diversity Foundation to carry out the fieldwork.

**Institutional Review Board Statement:** The survey met the standard of CNIL (2018) [54], the National Commission for Information Technology and Civil Liberties, and therefore did not require ethical approval.

**Informed Consent Statement:** Informed consent was obtained from all subjects involved in the study.

**Data Availability Statement:** Not applicable.

**Acknowledgments:** We acknowledge the effort by the local community to support the interviews and conservation actions we are carrying out in Cajón del Maipo. We also appreciate the assistance and participation of the practitioners from the Santuario de la Naturaleza Cascada de las Ánimas and the Santuario de la Naturaleza Lagunillas. Finally, MG-G acknowledges the theoretical help from the Nucleo de Estudios Sistémicos Transdisciplinarios for providing experience in social science methodological and epistemological tools.

**Conflicts of Interest:** The authors declare no conflict of interest.

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
