# Peer review of "Local Territorial Practices Inform Co-Production of a Rewilding Project in the Chilean Andes"

_sustainability, doi:10.3390/su15075966_

Round 1

Reviewer 1 Report

I think the discussion maybe is somewhat too long, it could be more stringent. On the other hand, it is an interesting discussion, so my suggestion is that the authors should look it over again and see if it is possible to concentrate some parts of the discussion. 

On line 239 starting with sheep and providing touristic services...., and then in the end of the line comes " ride no horses" , I believe it should be "ride horses" to specific places of the mountain.

Author Response

As the reviewer suggested, we work on synthesizing parts of the discussion to make it more precise.

Additionally, we changed "ride no horses" by "ride horses" in line 239.

Reviewer 2 Report

Please, find my remarks on the .pdf file.

Author Response

Line 156. We changed it as suggested.

Line 178. We changed it as suggested

Line 180. We think the concept is well-used

Lines 196-197. We added more details and we referenced the specific cite for more details.

Line 239. We changed it as suggested.

Line 418. We changed it and added reasons for opposing to the projects rather than reasons for not working (included changes in line 437).

Line 447. We changed the paragraph to the next section.

Line 510. We added mor details about the dogs.

Line 528. We added more information about the espinal habitat.

Line 565. We chaged it as suggested.

Line 585-586. We added more information to be clear about the analysis we showed in figure 3.

Line 686-687. We do not agree. This statement do not support the idea that “pumas would have been introduced on the area”. Rather, it support the idea that they perceive different kinds of pumas as a consequence of the perception of reintroduction by SAG.

Line 712. We agree with the commentary but is not the focus of those lines.

Reviewer 3 Report

This study, which includes local regional practices related to a rewilding project in the Chilean Andes, has been defined and examined according to the previous and current theoretical background and scientific research on the subject. I think the research design, questions, hypotheses and methods are clearly defined. The arguments defended were supported by the method followed and the findings were discussed in a consistent, balanced and persuasive manner. English language and style in some parts of the article require minor spelling, so authors should review the article thoroughly to make these minor corrections.

Author Response

Thank you very much. 

We checked the english and corrected some minor errors.